# The Effectiveness of an Evidence-Based Practice (EBP) Educational Program on Undergraduate Nursing Students’ EBP Knowledge and Skills: A Cluster Randomized Control Trial

**DOI:** 10.3390/ijerph18010293

**Published:** 2021-01-03

**Authors:** Daniela Cardoso, Filipa Couto, Ana Filipa Cardoso, Elzbieta Bobrowicz-Campos, Luísa Santos, Rogério Rodrigues, Verónica Coutinho, Daniela Pinto, Mary-Anne Ramis, Manuel Alves Rodrigues, João Apóstolo

**Affiliations:** 1Health Sciences Research Unit: Nursing, Nursing School of Coimbra, Portugal Centre for Evidence-Based Practice: A Joanna Briggs Institute Centre of Excellence, 3004-011 Coimbra, Portugal; fcardoso@esenfc.pt (A.F.C.); rogerio@esenfc.pt (R.R.); demar7@gmail.com (M.A.R.); apostolo@esenfc.pt (J.A.); 2FMUC—Faculty of Medicine, University of Coimbra, 3000-370 Coimbra, Portugal; 3Alfena Hospital—Trofa Health Group, Health Sciences Research Unit: Nursing, Nursing School of Coimbra, 3000-232 Coimbra, Portugal; filipadccouto@gmail.com; 4Health Sciences Research Unit: Nursing, Nursing School of Coimbra, 3004-011 Coimbra, Portugal; elzbieta.campos@gmail.com (E.B.-C.); luisasants@esenfc.pt (L.S.); vcoutinho@esenfc.pt (V.C.); danielapinto@esenfc.pt (D.P.); 5Mater Health, Evidence in Practice Unit & Queensland Centre for Evidence Based Nursing and Midwifery: A Joanna Briggs Institute Centre of Excellence, 4101 Brisbane, Australia; Mary-Anne.Ramis5@mater.org.au

**Keywords:** education, curriculum, education, nursing, evidence-based practice, knowledge, nursing education research, students, nursing

## Abstract

Evidence-based practice (EBP) prevents unsafe/inefficient practices and improves healthcare quality, but its implementation is challenging due to research and practice gaps. A focused educational program can assist future nurses to minimize these gaps. This study aims to assess the effectiveness of an EBP educational program on undergraduate nursing students’ EBP knowledge and skills. A cluster randomized controlled trial was undertaken. Six optional courses in the Bachelor of Nursing final year were randomly assigned to the experimental (EBP educational program) or control group. Nursing students’ EBP knowledge and skills were measured at baseline and post-intervention. A qualitative analysis of 18 students’ final written work was also performed. Results show a statistically significant interaction between the intervention and time on EBP knowledge and skills (*p* = 0.002). From pre- to post-intervention, students’ knowledge and skills on EBP improved in both groups (intervention group: *p* < 0.001; control group: *p* < 0.001). At the post-intervention, there was a statistically significant difference in EBP knowledge and skills between intervention and control groups (*p* = 0.011). Students in the intervention group presented monographs with clearer review questions, inclusion/exclusion criteria, and methodology compared to students in the control group. The EBP educational program showed a potential to promote the EBP knowledge and skills of future nurses.

## 1. Introduction

Evidence-based practice (EBP) is defined as “clinical decision-making that considers the best available evidence; the context in which the care is delivered; client preference; and the professional judgment of the health professional” [1] (p. 2). EBP implementation is recommended in clinical settings [2,3,4,5] as it has been attributed to promoting high-value health care, improving the patient experience and health outcomes, as well as reducing health care costs [6]. Nevertheless, EBP is not the standard of care globally [7,8,9], and some studies acknowledge education as an approach to promote EBP adoption, implementation, and sustainment [10,11,12,13,14,15].

It has been recommended that educational curricula for health students should be based on the five steps of EBP in order to support developing knowledge, skills, and positive attitudes toward EBP [16]. These steps are: translation of uncertainty into an answerable question; search for and retrieval of evidence; critical appraisal of evidence for validity and clinical importance; application of appraised evidence to practice; and evaluation of performance [16].

To respond to this recommendation, undergraduate nursing curricula should include courses, teaching strategies, and training that focus on the development of research and EBP skills for nurses to be able to incorporate valid and relevant research findings in practice. Nevertheless, teaching research and EBP to undergraduate nursing students is a challenging task. Some studies report that undergraduate students have negative attitudes/beliefs toward research and EBP, especially toward the statistical components of the research courses and the complex terminology used. Additionally, students may not understand the importance of the link between research and clinical practice [17,18,19]. In fact, a lack of EBP and research knowledge is commonly reported by nurses and nursing students as a barrier to EBP. It is imperative to provide the future nurses with research and EBP skills in order to overcome the barriers to EBP use in clinical settings.

At an international level, several studies have been performed with undergraduate nursing students to assess the effectiveness of EBP interventions on multiple outcomes, such as EBP knowledge and skills [20,21,22,23]. The Classification Rubric for EBP Assessment Tools in Education (CREATE) [24] suggests EBP knowledge should be assessed cognitively using paper and pencil tests, as EBP knowledge is defined as “learners’ retention of facts and concepts about EBP” [24] (p. 5). Additionally, the CREATE framework suggests EBP skills should be assessed using performance tests, as skills are defined as “the application of knowledge” [24] (p. 5). Despite these recommendations, few studies have assessed EBP knowledge and skills using both cognitive and performance instruments.

Therefore, this study aims to evaluate the effectiveness of an EBP educational program on undergraduate nursing students’ EBP knowledge and skills using a specific cognitive and performance instrument. The intervention used in this study was recently developed [25], and this is the first study designed to assess its effectiveness in undergraduate EBP.

## 2. Materials and Methods 

### 2.1. Design

A cluster randomized controlled trial with two-armed parallel group design was undertaken (ClinicalTrials.gov Identifier: NCT03411668).

### 2.2. Sample Size Calculation

The sample size was calculated using the software G*Power 3.1.9.2. (Heinrich-Heine-Universität Dusseldorf, Düsseldorf, Germany) Recognizing that there were no studies performed a priori using a cognitive and performance instrument to assess the effectiveness of an EBP educational program on undergraduate nursing students’ EBP knowledge and skills, we used an effect size of 0.25, which is a small effect size as proposed by Cohen [26]. A power analysis based on a type I error of 0.05; power of 0.80; effect size *f* = 0.25; and ANOVA repeated measures between factors determined a sample size of 98 as total.

Taking into account that our study used clusters (optional courses) and that each one had an average of 25 students, we needed at least four clusters to cover the total sample size of 98. However, to cover potential losses to follow-up, we included a total of six optional courses.

### 2.3. Participants’ Recruitment and Randomization

We recruited participants from one Portuguese nursing school in 2018. From the 12 optional clinical nursing courses (such as Community Nursing Intervention in Vulnerable Groups; Ageing; Health and Citizenship; The Child with Special Needs: Diagnoses and Interventions in Pediatric Nursing; Liaison Psychiatry Nursing; Nursing in the Emergency Room; etc.) in the 8th semester of the nursing program (last year before graduation), students from three clinical nursing courses were randomly assigned to the experimental group (EBP educational program) and students from another three clinical nursing courses were randomly assigned to the control group (no intervention—*education as usual*) before the baseline assessment. An independent researcher performed this assignment using a random number generator from the random.org website [27]. This assignment was performed based on a list of the 12 optional courses provided through the nursing school’s website.

### 2.4. Intervention Condition

The participants in the intervention group received education as usual plus the EBP educational program, which was developed by Cardoso, Rodrigues, and Apóstolo [25]. This intervention included EBP contents regarding models of thinking about EBP, systematic reviews types, review question development, searching for studies, study selection process, data extraction, and data synthesis.

This program was implemented in 6 sessions over 17 weeks:Sessions 1–3—total of 12 h (4 h per session) during the first 7 weeks using expository methods with practice tasks to groups of 20–30 students.Sessions 4–6—total of 6 h (2 h per session) during the last 10 weeks using active methods through mentoring to groups of 2–3 students.

Due to the nature of the intervention, it was not possible to blind participants regarding treatment assignment nor was it feasible to blind the individuals delivering treatment.

### 2.5. Control Condition

The participants in the control group received only education as usual; i.e., students allocated to this control condition received the standard educational contents (theoretical, theoretical–practical, practical) delivered by the nursing educators of the selected nursing school.

### 2.6. Assessment

All participants were assessed before (week 0) and after the intervention (week 18) using a self-report instrument. EBP knowledge and skills were assessed by the Adapted Fresno Test for undergraduate nursing students [28]. This instrument was adapted from the Fresno Test, which was originally developed in 2003 to measure knowledge and skills on EBP in family practice residents [29]. The Adapted Fresno Test for undergraduate nursing students has seven short answer questions and two fill-in-the-blank questions [28]. At the beginning of the instrument, two scenarios, which suggest clinical uncertainty, are presented. These two scenarios are used to guide the answers to questions 1 to 4: (1) write a clinical question; (2) identify and discuss the strengths and weaknesses of information sources as well as the advantages and disadvantages of information sources; (3) identify the type of study most suitable for answering the question of one of the clinical scenarios and justify the choice; and (4) describe a possible search strategy in Medline for one of the clinical scenarios, explaining the rationale. The next three short answer questions require that the students identify topics for determining the relevance and validity of a research study and address the magnitude and value of research findings. The last two questions are fill-in-the-blank questions. The answers are scored using a modified standardized grading system [28], which was adapted from the original [29]. The instrument has a total minimum score of 0 and a maximum score of 101. The inter-rater correlation for the total score of the Adapted Fresno Test was 0.826 [28]. The rater that graded the answers to the Adapted Fresno Test was blinded to treatment assignment.

Despite the fact that in the study proposal we did not consider any kind of qualitative analysis in order to assess EBP knowledge and skills in a more practical context, we decided during the development of the study to perform a qualitative analysis of monographs at the posttest. The monographs were developed by small groups of nursing students and were the final written work submitted by the students for their bachelor’s degree course. In this work, the students were asked to define a review question regarding the context of clinical practice where they were performing their clinical training. Students then proceeded to answer the review question through a systematic process of searching and selecting relevant studies and extracting and synthesizing the data. From the 58 submitted monographs (30 from the control group and 28 from the intervention group), 18 were randomized for evaluation (nine from the control group and nine from the intervention group) by an independent researcher using the random.org website [27] based on a list provided by the research team. Three independent experts (one psychologist with a doctoral qualification and two qualified nurses, one with a master’s degree) performed a qualitative analysis of the selected monographs. All experts had experience with the EBP approach and were blinded to treatment assignment. The experts independently used an evaluation form to guide the qualitative analysis of each monograph. This form presented 11 guiding criteria regarding review questions, inclusion/exclusion criteria, methodology (namely search strategy, study selection process, data extraction, and data synthesis), results presentation, and congruency between the review questions and the answers to them that were provided in the conclusion section. Thereafter, the experts met to discuss any discrepancies in their qualitative analysis until consensus was reached.

### 2.7. Statistical Analyses 

The data were analyzed using Statistical Package for the Social Sciences (SPSS; version 24.0; SPSS Inc., Chicago, IL, USA). Differences in sociodemographic characteristics of study participants and outcome data at baseline were analyzed using Pearson’s chi-squared test for nominal data and independent the *t*-test for continuous data.

Taking into account the central limit theorem and that ANOVA tests are robust to violation of assumptions [30], we decided to perform two-way mixed ANOVA to compare the outcome between and within groups. The Wilcoxon signed-rank test was used to analyze how many participants had improved their EBP knowledge and skills item-by-item, how many remained the same, and how many had decreased performance within each group. Statistical significance was determined by *p*-values less than 0.05.

To minimize the noncompliance impact, an intention-to-treat (ITT) analysis was used to analyze participants in the groups that they were initially randomized to [31] by using the last observation carried forward imputation method.

### 2.8. Ethics

This study was approved by the Ethical Committee of the Faculty of Medicine of the University of Coimbra (Reference: CE-037/2017). The institution where the study was carried out provided written approval. All participants gave informed consent, and the data were managed in a confidential way.

## 3. Results

Twelve potential clusters (optional courses in the 8th semester of the nursing program) were identified as eligible for this study. Of these, three were randomized for the intervention group and three for the control group. During the intervention, eight participants (two in the intervention group and six in the control group) were lost to follow-up because they did not fill-in the instrument in the post-intervention. Figure 1 shows the flow of participants through each stage of the trial.

### 3.1. Demographic Characteristics

As Table 1 displays, 148 undergraduate nursing students with an average age of 21.95 years (SD = 2.25; range: 21–41) participated in the study. A large majority of the sample were female (*n* = 118, 79.7%), had a 12th grade educational level (*n* = 144, 97.3%), and had participated in some form of EBP training (*n* = 121, 81.8%).

At baseline, the experimental and control groups were comparable regarding sex, age, education, EBP training, and performance on the Adapted Fresno Test (Table 1 and Table 3). The baseline data were similar with dropouts excluded; therefore, only ITT analysis results are presented.

### 3.2. EBP Knowledge and Skills

#### 3.2.1. Adapted Fresno Test

The two-way mixed ANOVA showed a statistically significant interaction between the intervention and time on EBP knowledge and skills, *F* (1, 146) = 9.550, *p* = 0.002, partial η^2^ = 0.061 (Table 2). Excluding the dropouts, the two-way mixed ANOVA analysis was similar. Thus, only the ITT analysis results are presented.

To determine the difference between groups at baseline and post-intervention, two separate between-subjects ANOVAs (i.e., two separate one-way ANOVAs) were performed. At the pre-intervention, there was no statistically significant difference in EBP knowledge and skills between groups: *F* (1,146) = 0.221, *p* = 0.639, partial η^2^ = 0.002. At the post-intervention, there was a statistically significant difference in EBP knowledge and skills between groups: *F* (1,146) = 6.720, *p* = 0.011, partial η^2^ = 0.044 (Table 3).

To determine the differences within groups from the baseline to post-intervention, two separate within-subjects ANOVAs (repeated measures ANOVAs) were performed. There was a statistically significant effect of time on EBP knowledge and skills for the intervention group: *F* (1,73) = 53.028, *p* < 0.001, partial η^2^ = 0.421 and for the control group: *F* (1,73) = 13.832, *p* < 0.001, partial η^2^ = 0.159 (Table 3).

The results of repeated measures ANOVA and between-subjects ANOVA analysis are similar if we exclude the dropouts; therefore, only ITT analysis results are presented.

The results of the Wilcoxon signed-rank test for each item of the Adapted Fresno Test are presented in Table 4. The results of this analysis revealed that students in both the intervention and control groups significantly improved their knowledge and skills in writing a focused clinical question (Item 1) (intervention group: *Z* = −4.572, *p* < 0.000; control group: *Z* = −2.338, *p* = 0.019), in building a search strategy (item 3) (intervention group: *Z* = −4.740, *p* < 0.000; control group: *Z* = −4.757, *p* < 0.000), in identifying and justifying the study design most suitable for answering the question of one of the clinical scenarios (item 4) (intervention group: *Z* = −4.508, *p* < 0.000; control group: *Z* = −3.738, *p* < 0.000), and in describing the characteristics of a study to determine its relevance (item 5) (intervention group: *Z* = −2.699, *p* = 0.007; control group: *Z* = −1.980, *p* = 0.048).

The students in the control group significantly improved their knowledge and skills in describing the characteristics of a study to determine its validity (item 6) (*Z* = −2.714, *p* = 0.007). The students in the intervention group significantly improved their knowledge and skills in describing the characteristics of a study to determine its magnitude and significance (item 7) (*Z* = −2.543, *p* = 0.011). No other significant differences were detected.

The results of the within groups comparison with the Wilcoxon signed-rank test are similar if we exclude the dropouts; therefore, only ITT analysis results are presented.

#### 3.2.2. Qualitative Analysis of Monographs

Based on the experts’ consensus report of each monograph, the analysis of the intervention group monographs showed that the students’ groups clearly defined their review questions and inclusion/exclusion criteria. These groups of students effectively searched for studies using appropriate databases, keywords, Boolean operators, and truncation. Additionally, we found thorough descriptions from students concerning the selection process, data extraction, and data synthesis. However, only three students’ groups provided a good description of the review findings with an appropriate data synthesis as well as a clear answer to the review question in the conclusion section of their monographs. It is noted that the criteria for the results and conclusion sections were more difficult to successfully achieve, even in the intervention group.

The monographs of the control groups showed weaknesses throughout. From the nine monographs of the control group, only two presented the review question in a way that was clearly defined. In all of the monographs, the inclusion/exclusion criteria were either not very informative, unclear, or did not match with the defined review questions. Additionally, the search strategies were not clear and demonstrated limited understanding, such as lack of use of appropriate synonyms, absent truncations, and no definition of the search field for each word or expression to be searched. None of the monographs from the control group reported information about the methods used to study the selection process, to extract data, or to synthesize data. In the conclusion section, students from the control group also demonstrated difficulties in synthesizing the data and limitations by providing a clear answer to the review question.

## 4. Discussion

This study sought to evaluate the effectiveness of an EBP educational program on undergraduate nursing students’ EBP knowledge and skills. Even though both groups improved after the intervention in EBP knowledge and skills, the study results showed that the improvement was greater in the intervention group. This result was reinforced by the results of the qualitative analysis of monographs.

To the best of our knowledge, this is the first study to use a cognitive and performance assessment instrument (Adapted Fresno Test) with undergraduate nursing students, as suggested by CREATE [24]. Additionally, it is the first study conducted using the EBP education program [25]. Therefore, comparison of our findings with similar studies in terms of the type of assessment instrument and intervention is limited.

However, comparing our study with other previous research using other types of instruments and interventions demonstrates similar results [20,21,22,23]. In a quasi-experimental study [20], it was found that an EBP educational teaching strategy showed positive results in improving EBP knowledge in undergraduate nursing students. A study showed that undergraduate nursing students who received an EBP-focused interactive teaching intervention improved their EBP knowledge [21]. Another study indicated that a 15-week educational intervention in undergraduate nursing students (second- and third-year) significantly improved their EBP knowledge and skills [22]. In addition, a study by Zhang, Zeng, Chen, and Li revealed a significant improvement in undergraduate nursing students’ EBP knowledge after participating in a two-phase intervention: a self-directed learning process and a workshop for critical appraisal of literature [23].

Despite the effectiveness of the program in improving EBP knowledge and skills, the students included in the present study had low levels of EBP knowledge and skills as assessed by the Adapted Fresno Test at the pretest and posttest. These low levels of EBP knowledge and skills, especially at the pretest, might have influenced our study results. As a matter of fact, the Adapted Fresno Test is a demanding test since it requires that students retrieve and apply knowledge while doing a task associated with EBP based on scenarios involving clinical uncertainty. Consequently, this kind of test is very useful to truly assess EBP knowledge retention and abilities in clinical scenarios that do not allow guessing the answers. Notwithstanding, due to these characteristics, the Adapted Fresno Test may possibly be less sensitive when small changes occur or when students have low levels of EBP knowledge and skills. Nevertheless, even using instruments with Likert scales, other studies also showed that students have low levels of EBP knowledge and skills [21,22,23].

The low levels of EBP knowledge and skills of the undergraduate nursing students may be a reflection of a persistent, traditional education with regard to research. By this we mean that the focus of training remains on primary research—preparing students to be “research generators” instead of preparing them to be “evidence users” [32]. Furthermore, the designed and tested intervention used in this study was limited in time (only 17 weeks), was provided by only two instructors, and was delivered to fourth-year undergraduate nursing students, which are limitations for curriculum-wide integration of EBP.

Indeed, a curriculum that promotes EBP should facilitate students’ acquisition of EBP knowledge and skills over time and with levels of increasing complexity through their participation in EBP courses and during their clinical practice experiences [32,33,34,35]. As Moch, Cronje, and Branson suggest, “It is only in such practical settings that students can experience the challenges intrinsic to applying scientific evidence to the care of real patients. In these clinical settings, students can experience both the frustrations and the triumphs inevitable to integrating scientific knowledge into patient care.” [35] (p. 11). Therefore, in future studies, other broad approaches for curriculum-wide integration of EBP as well as its long-term effects should be evaluated.

Previously in the Discussion, we highlighted the limitations of the proposed intervention in terms of time constraints (only 17 weeks), instructors’ constraints (only two instructors provided the intervention), and participants’ constraints (fourth-year undergraduate nursing students). In addition, the study was also restricted to one Portuguese nursing school, which can limit the generalization of the results. However, our study tried to address some of the fragilities identified in other studies [20,21,22,23] on the effectiveness of EBP educational interventions by including a control group and by measuring EBP knowledge and skills with an objective measure and not a self-reported measure.

Bearing this in mind, future studies in multiple sites should assess the long-term effects of the EBP educational intervention and the impact on EBP knowledge and skills of potential variations in contents and teaching methods. In addition, studies using more broad interventions for curriculum-wide integration of EBP should also be performed.

## 5. Conclusions

Our findings show that the EBP educational program was effective in improving the EBP knowledge and skills of undergraduate nursing students. Therefore, the use of an EBP approach as a complement to the research education of undergraduate nursing students should be promoted by nursing schools and educators. This will help to prepare the future nurses with the EBP knowledge and skills that are essential to overcome the barriers to EBP use in clinical settings, and consequently, to contribute to better health outcomes.

## Figures and Tables

**Figure 1 ijerph-18-00293-f001:**
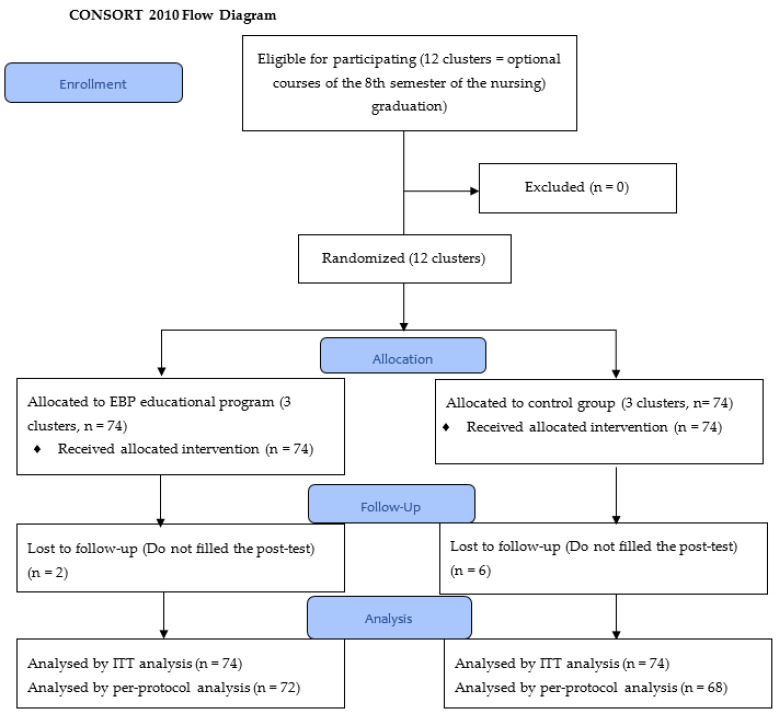
Consolidated Standards of Reporting Trials (CONSORT) diagram showing the flow of participants through each stage of the trial. ITT: intention-to-treat.

**Table 1 ijerph-18-00293-t001:** Socio-demographic characterization of the sample—ITT analysis.

	Total	Intervention Group	Control Group		
	(*n* = 148)	(*n* = 74)	(*n* = 74)		
	Mean ± SD	Mean ± SD	Mean ± SD	Independent *t*-test	*p*-Value *
	(Min–Max)	(Min–Max)	(Min–Max)		
Age in years	21.95 ± 2.25	22.20 ± 2.84	21.70 ± 1.42	1.353	0.178
	(21–41)	(21–41)	(21–31)		
	***n* (%)**	***n* (%)**	***n* (%)**	***X^2^***	***p*-Value ***
Female	118 (79.7)	63 (85.1)	55 (74.3)	2.676	0.102
Male	30 (20.3)	11 (14.9)	19 (25.7)		
Education				0.993	0.609
12th grade	144 (97.3)	72 (97.3)	72 (97.3)		
Graduation	2 (1.4)	1 (1.4)	1 (1.4)		
Master	1 (0.7)	1 (1.4)	-		
Missing	1 (0.7)	-	1 (1.4)		
EBP training *				0.221	0.638
Yes	121 (81.8)	59 (79.7)	62 (83.8)		
No	26 (17.6)	14 (18.9)	12 (16.2)		
Missing	1 (0.7)	1 (1.4)	-		

* Defined as any kind and duration of evidence-based practice (EBP) training, such as EBP contents in a course, a workshop, a seminar.

**Table 2 ijerph-18-00293-t002:** Main effects of time and group and interaction effects on EBP knowledge and skills—ITT analysis.

Outcome Measure	Effects	*F*	*p*-Value	Partial Eta^2^
EBP knowledge and skills assessed by Adapted Fresno Test	Time × Group	9.550	0.002	0.061

**Table 3 ijerph-18-00293-t003:** Repeated measures ANOVA and between-subjects ANOVA—ITT analysis.

		Baseline	Post-Test		
		Mean ± SD	Mean ± SD	Repeated Measures ANOVA	*p*
EBP knowledge and skills assessed by Adapted Fresno Test	intervention group (*n* = 74)	6.85 ± 5.16	12.47 ± 7.21	53.028	<0.001
	Control group (*n* = 74)	7.26 ± 5.34	9.73 ± 5.56	13.832	<0.001
Between-subjects ANOVA		0.221	6.720		
*p*		0.639	0.011		

**Table 4 ijerph-18-00293-t004:** Within groups comparison with Wilcoxon signed-rank test for each item of the Adapted Fresno Test—ITT analysis.

	Intervention Group (*n* = 74)				Control Group (*n* = 74)			
	Status	*n*	*Z*	*p*	Status	*n*	*Z*	*p*
Item 1	Improved	43	−4.572	<0.000	Improved	29	−2.338	0.019
	Decreased	13			Decreased	16		
	Maintained	18			Maintained	29		
Item 2	Improved	20	−1.498	0.134	Improved	24	−0.371	0.711
	Decreased	32			Decreased	19		
	Maintained	22			Maintained	31		
Item 3	Improved	49	−4.740	<0.000	Improved	41	−4.757	<0.000
	Decreased	14			Decreased	10		
	Maintained	11			Maintained	23		
Item 4	Improved	43	−4.508	<.000	Improved	33	−3.738	<.000
	Decreased	8			Decreased	10		
	Maintained	23			Maintained	31		
Item 5	Improved	9	−2.699	0.007	Improved	6	−1.980	0.048
	Decreased	0			Decreased	1		
	Maintained	65			Maintained	67		
Item 6	Improved	12	−1.236	0.216	Improved	4	−2.714	0.007
	Decreased	9			Decreased	15		
	Maintained	53			Maintained	55		
Item 7	Improved	11	−2.543	0.011	Improved	8	−1.941	0.052
	Decreased	2			Decreased	2		
	Maintained	61			Maintained	64		
Item 8	Improved	1	−0.577	0.564	Improved	2	−1.134	0.257
	Decreased	2			Decreased	5		
	Maintained	71			Maintained	67		
Item 9	Improved	4	−0.378	0.705	Improved	5	0.000	1.000
	Decreased	3			Decreased	5		
	Maintained	67			Maintained	64		
Total Adapted Fresno Test	Improved	54	–5.780	0.000	Improved	45	−3.354	0.001
	Decreased	13			Decreased	17		
	Maintained	7			Maintained	12		

## Data Availability

The data presented in this study are available on request from the corresponding author. The data are not publicly available because this issue was not considered within the informed consent signed by the participants of the study.

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
