# Peer review of "The Effectiveness of an Evidence-Based Practice (EBP) Educational Program on Undergraduate Nursing Students’ EBP Knowledge and Skills: A Cluster Randomized Control Trial"

_ijerph, 2021, doi:10.3390/ijerph18010293_

Round 1

Reviewer 1 Report

Thank you for the opportunity to review your interesting and relevant manuscript evaluating the implementation of EBP education to undergraduate students. As you stated, this is an area that needs additional research. Your team has done an outstanding job on this research. The inclusion of both qualitative and quantitative assessments strengthens your results. Just a few observations:

  1. Figure 1. Control group =recruited 74 students - 6 (lost to follow-up) = 68. The final n for the control group is 72 in Figure 1; unclear of the basis for 72.
  2. Table 1. EBP training. Curious as how you defined EBP training? # of hours? A project? A module in a course?
  3. So, if I am reading this correctly, the intervention group improved in 5 items on the Adapted Fresno Test and the control group improved in 6 items on the Adapted Fresno Test. Neither group improved on items 8 or 9. I am not sure that this would strongly support that the intervention was effective in improving EBP knowledge and skills.
  4. However, it appears the qualitative monographs showed a different result than the Adapted Fresno Test and supports the effectiveness of your intervention. I am wondering if the issue is the Adapted Fresno Test is the best option for evaluating EBP knowledge and skills.
  5. I would suggest that your discussion address this inconsistency in the measurements and the limitations of the Adapted Fresno Test.

Great work! Looking forward to seeing this in print.

Author Response

Dear Reviewer,

Thank you so much for your careful analysis of the paper and valuable comments. They help us to improve and clarify some parts of the study that I think that will help the readers to understand and analyze our design and results.

Please see below the answers and pages where we change or add information according to each of your concern.

Reviewer Comments

Answer to reviewer

Page

1.    Figure 1. Control group =recruited 74 students - 6 (lost to follow-up) = 68. The final n for the control group is 72 in Figure 1; unclear of the basis for 72.

Thank you for noted. It was a mistake. As we used the Intention-to-treat analysis, we need to consider here a n= 74 in both groups. Please see our corrections and additions in order to clarify.

5

2.    Table 1. EBP training. Curious as how you defined EBP training? # of hours? A project? A module in a course?

We accepted any kind and duration of EBP training, such as EBP contents in a course, a workshop, a seminar. To clarify, we added this information as note to the table 1.

6

3.    So, if I am reading this correctly, the intervention group improved in 5 items on the Adapted Fresno Test and the control group improved in 6 items on the Adapted Fresno Test. Neither group improved on items 8 or 9. I am not sure that this would strongly support that the intervention was effective in improving EBP knowledge

4.    However, it appears the qualitative monographs showed a different result than the Adapted Fresno Test and supports the effectiveness of your intervention. I am wondering if the issue is the Adapted Fresno Test is the best option for evaluating EBP knowledge and skills.and skills.

If we look only for the results showed by the Wilcoxon signed-rank test, indeed both groups improved after the intervention considering the Total Adapted Fresno Test and also the individual items. 

Moreover, if we analyze the Repeated measures ANOVA (Table 3), we also noted that both groups improved after the interventions. However, when we analysis the results of Between-subjects ANOVA (Table 3), we see that despite the groups are comparable at baseline, at post-test the groups presented a statistically significant difference. This graph that we did not include in the paper also highlight this difference.

In order to point this out, we rephrase a phrase on discussion section (lines 295-299)

8

5.    I would suggest that your discussion address this inconsistency in the measurements and the limitations of the Adapted Fresno Test.

We added in lines 318—326 information in order to provide some limitations of the Adapted Fresno Test. As we explain in point before, we added a phrase (lines 295-296) regarding the assessment using Adapted Fresno Test and qualitative analysis of monographs.

8

Reviewer 2 Report

This paper addresses a relevant and interesting topic of research to assess the effectiveness of an educational intervention in undergraduate nursing students to enhance EBP knowledge and skills with a cluster randomized controlled trial design.

In general, the paper is very well written. The authors highlight the importance of the health sciences curricula to include EBP courses designed for the development of the skills to search, evaluate and incorporate valid evidence to daily practice. The study is well identified as a cluster randomised trial as well as the eligibility criteria for clusters. The primary outcome pertains to the cluster level and is focused on EBP knowledge and skills.

In the methods section it is described as a parallel group design, sample sizes are well calculated (number of students per cluster and number of clusters) including the reasons to increase the number of clusters to achieve the required power. Eligibility criteria for participants and clusters is detailed, as well for locations where the data were collected. Do the randomization was after or before the baseline assessment? Was the same person who enrolled clusters and assigned clusters to interventions?  It is not mentioned which courses were taken by the control group, they only say “another three courses” this is confusing for the reader because they could be EBP courses or another subject not-EBP related courses (this would explain partially the results when both groups showed improvement in knowledge and skills, if not should explain). Overall, the intervention is well described with details to allow replication, but they are short when they stand that the control group had education as usual (what is usual means?). 

Was the last part of qualitative analysis included in the study proposal? If it was an additional analysis given the primary results it should be noted. I suggest having a detailed description of the way the monographs were randomized and the process of grading or evaluate objectively the performance in search strategies, synthesize and conclude.

Results are described at the cluster level and a flow diagram as recommended. They present a table with baseline demographic characteristics by group. They also present a table for the primary outcome (knowledge & skills) for each group at baseline and post test with between-subjects ANOVA with ITT, and a table to compare within groups for each item of the instrument. Results of the qualitative analysis of monographs are presented, and would be beneficial if they describe if they used a checklist or how they concluded that the intervention group monographs had better overall performance.

Results are discussed and some limitations are addressed. I suggest adding their insights about the generalisability of their results, and where the full trial protocol can be accessed to consider replicability of the study in other contexts. Funding and conflict of interests information is included.

Findings are important input for the nursing curriculum academia as they are probing the effectiveness of the EBP educational program (18 hours-6 sessions) to improve EBP knowledge and skills of the undergraduate nursing students.

Author Response

Dear Reviewer,

Thank you so much for your deeply analysis and valued comments. We addressed them and we think that now the methodology as well as the results/discussion are much better clear for an external reader.

Please see below the answers and pages where we change or add information according to each of your concern.

Reviewer comment

Answer

Page

Do the randomization was after or before the baseline assessment? Was the same person who enrolled clusters and assigned clusters to interventions? 

The randomization was done before the baseline assessment.

As we stated at lines 104-105, the person that done the assignment was independent to the study, meaning that he did not know the participants.

This researcher performed the assignment based on the list of optional courses stated at Nursing School website. We added this information to clarify.

3

It is not mentioned which courses were taken by the control group, they only say “another three courses” this is confusing for the reader because they could be EBP courses or another subject not-EBP related courses (this would explain partially the results when both groups showed improvement in knowledge and skills, if not should explain).

In fact, the 12 optional courses are the following clinical nursing courses:

1.     Community Nursing Intervention in Vulnerable Groups

2.     School-based Community Nursing Intervention

3.     Community Mental Health and Rehabilitation Nursing

4.     Nursing and Sexual and Reproductive Health

5.     Ageing, Health and Citizenship      

6.     The child with special needs: Diagnoses and interventions in pediatric nursing         

7.     Self-Care in Situations of Dependency

8.     Liaison Psychiatry Nursing

9.     Management of Therapeutic Self-Care

10.  Nursing in the Emergency Room

11.  Perioperative Nursing

12.  Nursing in Intensive Care

To clarify, we provided some additional information on this, including some examples of the courses.

Page 3

Overall, the intervention is well described with details to allow replication, but they are short when they stand that the control group had education as usual (what is usual means?). 

To clarify this point, we added a subpoint related to the control condition – education as usual. Moreover, in the intervention condition, we clarify that the intervention group received the education as usual plus the EBP educational program.

Page 3

Was the last part of qualitative analysis included in the study proposal? If it was an additional analysis given the primary results it should be noted. I suggest having a detailed description of the way the monographs were randomized and the process of grading or evaluate objectively the performance in search strategies, synthesize and conclude.

Indeed, the qualitative analysis was not included in the study proposal. However, when we started the intervention, we considered that this kind of qualitative data would also be significant for our analysis as a complement to the objective measures. We made some changes to clarify this on lines 148-150.

In lines 158-159, we added information regarding the randomization of monographs.

The analysis of monographs was made through a qualitative analysis. We do not used a grading system. The three experts reach a consensus on the qualitative analyses that they performed. Please see our changes in lines 162-167.

Page 4

Results of the qualitative analysis of monographs are presented, and would be beneficial if they describe if they used a checklist or how they concluded that the intervention group monographs had better overall performance.

As we clarify in the Material and Methods section (lines 162-167), it was not used a checklist that allows an objective assessment. This analysis was based on the qualitative report of the experts.

Page 8

I suggest adding their insights about the generalisability of their results, and where the full trial protocol can be accessed to consider replicability of the study in other contexts.

As we cited in lines 87-88, the trial was register in ClinicalTrials.gov (Identifier: NCT03411668), so there you can see the registration.

We added some insights about generalizability of the results in lines 342-349

Page 2

Page